# Predictive Value of Circulating miRNAs in Lymph Node Metastasis for Colon Cancer

**DOI:** 10.3390/genes12020176

**Published:** 2021-01-27

**Authors:** In Hee Lee, Gyeonghwa Kim, Sang Gyu Kwak, Dong Won Baek, Byung Woog Kang, Hye Jin Kim, Su yeon Park, Jun Seok Park, Gyu-Seog Choi, Keun Hur, Jong Gwang Kim

**Affiliations:** 1Division of Hematology/Oncology, Department of Internal Medicine, Daegu Catholic University School of Medicine, Daegu 42112, Korea; cakey83@hanmail.net; 2Department of Biochemistry and Cell Biology, School of Medicine, Kyungpook National University, Daegu 41404, Korea; med.aurora1106@gamil.com; 3Department of Medical Statistics, Daegu Catholic University School of Medicine, Daegu 42112, Korea; sanggyu39@naver.com; 4Department of Oncology/Hematology, Kyungpook National University Chilgok Hospital, School of Medicine, Kyungpook National University Cancer Research Institute, Kyungpook National University, Daegu 41404, Korea; baekdw83@gamil.com (D.W.B.); bwkang@knu.ac.kr (B.W.K.); 5Department of Surgery, Kyungpook National University Chilgok Hospital, School of Medicine, Kyungpook National University Cancer Research Institute, Kyungpook National University, Daegu 41404, Korea; chocogom@hanmail.net (H.J.K.); psy-flower@hanmail.net (S.y.P.); caumed5@hanmail.net (J.S.P.); kyuschoi@knu.ac.kr (G.-S.C.)

**Keywords:** microRNA, biomarker, circulating microRNAs, lymph node metastasis, colon

## Abstract

(1) Background: Lymph node (LN) status is an indubitable prognostic factor for survival among colon cancer patients. MicroRNAs (miRNAs) have been implicated in the development and progression of many cancers and are potential biomarkers for cancer diagnosis and prognosis. Therefore, we validated candidate biomarkers using circulating miRNAs by analyzing the plasma miRNA concentrations from patients with colon cancer to predict LN metastasis. (2) Methods: This study included 79 blood samples from patients diagnosed with colon cancer. The NanoString assay was used for screening, and TaqMan miRNA assays for quantitative real-time polymerase chain reaction (RT-PCR) test was used for validation. In a discovery set, we compared the expression of 800 circulating miRNAs in 24 samples (stage 0/I/IIA versus IIIB/IIIC). For validation, a total 79 samples were tested using quantitative RT-PCR. (3) Results: In the discovery set, 10 candidate circulating miRNAs were detected (4 up-regulated miRNAs: miR-323a-3p, miR-382-5p, miR-29a-3p, and miR-376a-3p; 6 down-regulated miRNAs: miR-26a-5p, let-7g-5p, miR-15b-5p, miR-142-3p, miR-374a-5p, and let-7b-5p). In the validation set, higher expression of three circulating miRNAs (miR-323a-3p, miR-382-5p, and miR-376a-3p) was significantly associated with LN metastasis (*p* = 0.0063, 0.0107, and 0.0022). (4) Conclusions: High expression of circulating miR-323a-3p, miR-382-5p, and miR-376a-3p was significantly associated with LN metastasis in colon cancer patients. These miRNAs could be circulating biomarker candidates that predict the presence of LN metastasis.

## 1. Introduction

The lymph node metastasis is a crucial factor for predicting disease recurrence and survival in patients with colon cancer. Compared to patients with stage II colon cancer, those with stage III colon cancer who have LN metastases have >20% lower 5-year survival rate (82.5% vs. 59.5%) [1]. Accordingly, LN status is an important factor for determining the administration of adjuvant chemotherapy after surgical resection [2,3,4].

The presence of LN metastases is an indubitable prognostic factor for colon cancer. LN metastases are a precursor of distant metastasis, and resection of regional LN is necessary to reduce the recurrence rates [1]. It is suggested that tumor cells spread from the primary tumor site to the lymph nodes via lymphatic vessels and consequently, to the next distant organ. Therefore, regional lymph node metastasis is believed an essential step in tumor cell dissemination in colorectal cancer. However, some colon cancer patients with a high T stage do not exhibit LN metastasis, whereas others have LN metastasis even during early T stage. Furthermore, 25% of LN negative patients experience recurrence and not all patients with positive LN have a poor prognosis [5]. Therefore, predictive biomarkers complementing the TNM classification are needed.

MicroRNAs (miRNAs) are a class of small noncoding RNAs that post-transcriptionally regulate the gene expression and are involved in a wide range of physiological and pathological processes [6]. Moreover, the dysregulation of miRNAs can play an essential role in tumor development and progression, and serve as a potential biomarker for cancer diagnosis and prognosis [7]. MiRNAs from different cell types can be secreted into the extracellular space and then transported to the circulating blood, such as the peripheral blood. These circulating miRNAs are highly stable in body fluids, and thus, these small molecules are a good predictive biomarker candidate for liquid biopsy [8]. A large number of studies have suggested the role of tissue miRNA as a biomarker for colon cancer diagnosis [9,10]. Feng et al. showed that the expression level of miRNA-141 in tumor tissues and LN were significantly decreased in colorectal cancer patients, with a more evident decline in cases with LN metastasis [11]. Drusco et al. identified that miRNA-21 was significantly overexpressed in metastatic LN than in LN-negative colon cancer [5]. Recently, using the TCGA database and integrated bioinformatics analysis, 73 differently expressed miRNAs associated with LN metastasis in colorectal cancer were identified [12]. However, there are limited data available on the comprehensive analysis of circulating miRNAs in LN metastasis in patients with colon cancer. Here, we identified circulating miRNA expression in patients with colon cancer with or without LN metastasis to evaluate the predictive biomarkers for LN metastasis.

## 2. Materials and Methods

### 2.1. Study Population and Sample Collection

Seventy-nine patients who underwent surgical excision for colon cancer without neoadjuvant treatment prior to surgery at the Kyungpook National University Chilgok Hospital (KNUCH) between January 2017 and February 2019 were enrolled in this study.

Blood samples were taken from patients the day before surgery. Plasma was prepared within 4 h after the blood draw by retaining the supernatant after centrifugation (2500× *g*; 15 min) and was stored in aliquots at −80 °C. Two separate patient cohorts were identified, a discovery set (*n* = 24) comprising 12 LN negative samples (stage 0/I/IIA) and 12 LN positive samples (stage IIIB/IIIC), and a validation set (*n* = 79) comprising 37 LN negative samples (stage 0/I/II) and 42 LN involved samples (stage III/IV). The study procedure was approved by the institutional review board at KNUCH (KNUCH 2019-10-025) and informed consent was obtained from all patients.

### 2.2. Circulating miRNA Extraction

We used the miRNeasy Serum/Plasma Kit (Qiagen Inc, Valencia, CA, USA) according to the manufacturer’s instructions to extract circulating miRNA from 79 plasma samples of colorectal cancer patients. The extraction of 79 circulating miRNAs were normalized by adding synthetic *Caenorhabditis. elegans* miR-39 (cel-miR-39, Applied Biosystems, Foster City, CA, USA) as *spike-in-control* to each plasma sample, as previously described [13]. The circulating miRNAs were determined using a NanoPhotometer N60 (Implen GmbH, München, Germany, USA).

### 2.3. miRNAs Expression Analysis

For discovery, we compared the expression of approximately 800 miRNAs using the Nanostring nCounter (NanoString Technologies, Seattle, WA, USA) gene expression technology in 24 plasma samples from colon cancer patients. The quantity and quality of the total RNAs were measured using a DS-11 spectrophotometer (Denovix, Wilmington, DE, USA) and Fragment Analyzer (Advanced Analytical Technologies, Inc., Ankeny, IA, USA). For validation, discovered miRNAs were investigated using TaqMan miRNA assays for quantitative real-time polymerase chain reaction (qRT-PCR) from a total of 79 plasma samples. The qRT-PCR values were normalized to synthetic *cel-miR-39*. The expression differences of circulating miRNAs were displayed using the 2^−^^ΔΔCt^ method. NanoString data were used to obtain the miRNA expression in each sample as a fold-change value, and a fold change of ±1.3 was considered significant.

### 2.4. Statistical Analyses

Comparisons of miRNA expression were assessed using a paired Student’s *t*-test and analysis of variance. Receiver operating characteristic (ROC) curves were constructed, and the area under the ROC curve (AUC) was calculated to evaluate the predictive performance of the miRNAs. Relapse-free survival (RFS) was estimated from the time of surgery until disease recurrence or death. Overall survival (OS) was calculated from the date of diagnosis to the time of death from any cause. Survival analysis was performed using the Kaplan–Meier method with a log-rank test. Multivariate analysis was performed using variables with a *p*-value of <0.1 in the univariate analysis using Cox’s proportional hazards model to derive a potentially suitable set of predictors. Two-sided *p*-values of <0.05 were considered significant. Statistical analyses were performed using SPSS software version 21.0 (SPSS, Inc., Chicago, IL, USA) and GraphPad Prism version 7.0 (GraphPad Software, San Diego, CA, USA).

## 3. Results

### 3.1. Patient Characteristics

The clinicopathologic characteristics of the patients in validation cohort are shown in Table 1. The median age of the patients was 63 years (range 37–86 years), and the ratio of males to females was 1.2:1. The pathologic stages were as follows: stage I (*n* = 9, 4.5%), stage II (*n* = 28, 35.4%), stage III (*n* = 39, 49.4%), and stage IV (*n* = 3, 3.8%). The number of median lymph node harvest in the resected specimens is 26 (range 13~79). All patients underwent R0 resection. Sixteen (20.3%) patients experienced relapse.

### 3.2. Discovery of Candidate miRNAs

Table 2 shows the results of discovery cohort. Among the 800 miRNAs examined, 10 miRNAs were found to be differentially expressed between the LN-negative (stage 0/I/IIA) and the LN-positive (stage IIIB/IIIC) groups in the Nanostring assay. Four were significantly upregulated (miR-323a-3p, miR-382-5p, miR-29a-3p, and miR-376a-3p) and six were significantly downregulated (miR-26a-5p, let-7g-5p, miR-15b-5p, miR-142-3p, miR-374a-5p, and let-7b-5p) in the LN-positive group than in the LN-negative group.

### 3.3. Validation of Differentially Expressed miRNAs Using Quantitative RT-PCR

First, the ten discovered miRNAs were quantitatively validated using qRT-PCR in 24 discovery cohort. Three up-regulated miRNAs (miR-323a-3p, miR-382-5p, and miR-376a-3p) in LN-positive group were successfully validated (Figure 1). Next, the three miRNAs (miR-323a-3p, miR-382-5p, and miR-376a-3p) were further validated in a validation cohort of 79 plasma samples using quantitative RT-PCR. All three miRNAs (miR-323a-3p, miR-382-5p, and miR-376a-3p) were significantly upregulated in the LN-positive group than in the LN-negative group (*p* = 0.0049, 0.0053, and 0.0024) (Figure 2).

### 3.4. Establishment and Validation of the 3-miRNAs Signature with Predictive Value

The predictive accuracy of these three miRNAs to distinguish LN positivity and LN negativity was measured by ROC curve analysis (Figure 3). To improve the predictive value of the miRNAs, a risk score method was applied to construct a signature combining the expression of these three miRNAs. Each patient was assigned a risk score that was calculated by logistic regression. The ROC curves of the miR-323a-3p, miR-382-5p, and miR-376a-3p levels for determining the presence of LN metastasis showed that miR-376a-3p had the highest AUC value as a single biomarker (Appendix A. AUC = 0.711). Furthermore, the 3-miRNA signature could predict LN metastasis better than the 1 or 2-miRNA signature, with an AUC of 0.799 (Figure 3a, AUC = 0.799).

### 3.5. Survival Analysis

There was no significant correlation between the expression of candidate miRNAs and survival in multivariate analysis (Table 3). However, RFS and OS curves using the 3-miRNA signature exhibited survival difference (Figure 4).

## 4. Discussion

The aim of this study was to investigate the potential role of circulating miRNAs as predictive biomarkers of LN metastasis in colon cancer patients. Based on the miRNA profiling followed by confirmation with quantitative RT-PCR, we identified three miRNAs (miR-323a-3p, miR-382-5p, and miR-376a-3p) significantly associated with LN metastasis in colon cancer patients. Furthermore, the 3-miRNA signature showed higher predictive value in identifying LN metastasis in colon cancer patients than each miRNA alone. However, the expression of these miRNAs was not associated with survival.

The biological functions of these three miRNAs have been previously investigated in some cancers. Ho et al. reported that miR-382-5p promotes breast cancer cell viability, clonogenicity, survival, migration, and invasion in vitro and breast tumorigenesis and metastasis in vivo. They reported that MCF-7 cells transfected with miR-382-5p exhibited a dose-dependent increase in migration and invasion compared to control groups. This miRNA functions as an oncomiR in breast cancer and is an independent predictor of higher incidence and poorer prognosis [14]. Furthermore, exosomal miR-382-5p overexpression is reportedly responsible for oral squamous cell carcinoma cell migration and invasion [15]. Therefore, miR-382-5p has an important role in cancer development and progression and may serve as a potential predictor for LN metastasis and prognosis in colon cancer.

However, less is known about miR-323a-3p and miR-376-3p. miR-323a-3p is known to enhance tumor angiogenesis in prostate cancer; thus, the upregulation of this miRNA contributes to an aggressive phenotype in prostate cancer [16]. Wu et al. found that miR-376-3p upregulation can affect the proliferation and migration of colon cancer cells [17]. However, there are fewer data regarding the predictive role of miR-323a-3p and miR-376-3p in colon cancer LN metastasis. In our study, we demonstrated that miR-323a-3p and miR-376-3p were significantly upregulated in the LN-positive group than in the LN-negative group (*p* = 0.0049, 0.0024, respectively).

Furthermore, we used the ROC curves of the three miRNA levels to determine the presence of LN metastasis and found that the combination of these three miRNAs was more effective than using them individually in the ROC analysis (Figure 3). To improve the predictive value of miRNAs, the method of summation of several miRNA levels had been applied in previous studies. Chen et al. validated a 4-microRNA signature to predict LN metastasis and prognosis in breast cancer by analyzing the ROC curves and risk score calculation [18].

In our study, we found three significantly upregulated miRNAs in the LN-positive group, although these miRNAs did not have impact on survival, despite the observable trend. There are still some limitations in our study. First, the sample size is limited. Moreover, short follow-up period can be the possible reason for the lack of statistically significant associations with survival. Furthermore, the exact function of miR-323a-3p, miR-382-5p, and miR-376a-3p in colon cancer remains to be clarified. In the future study, a functional study with a larger study samples will be required to confirm the present results.

In conclusion, colon cancer patients with LN metastasis had higher plasma levels of circulating miR-323a-3p, miR-382-5p, and miR-376a-3p than colon cancer patients without LN metastasis. These miRNAs have the potential to be circulating biomarkers to predict LN metastasis in colon cancer. Moreover, functional studies on miR-323a-3p, miR-382-5p, and miR-376a-3p in colon cancer are warranted.

## Figures and Tables

**Figure 1 genes-12-00176-f001:**
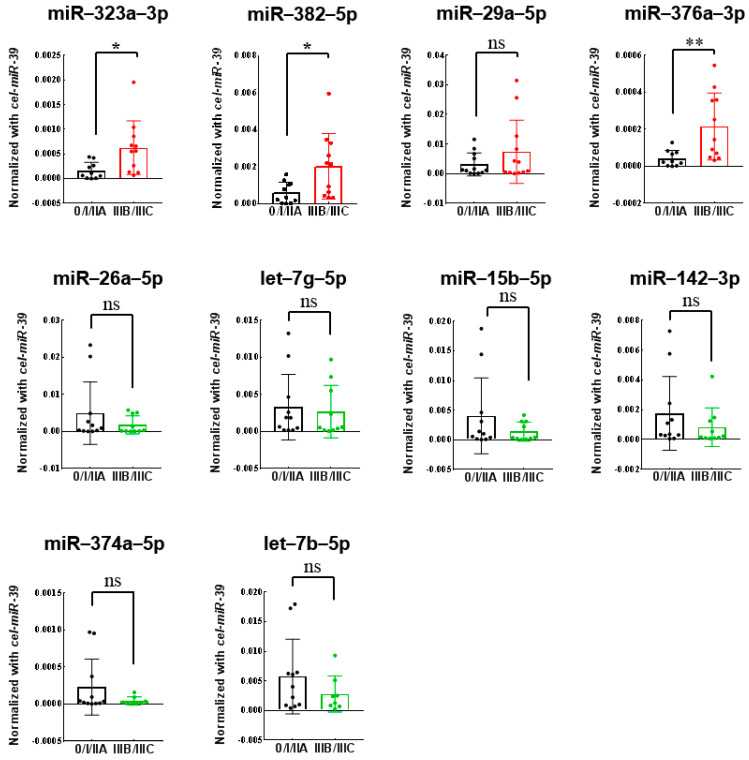
Analysis of 24 plasma samples from colon cancer patients using quantitative real-time polymerase chain reaction (RT-PCR) (discovery cohort, stage 0/I/IIA vs. IIIB/IIIC). miR-323a-3p (* *p* = 0.0177), miR-382-5p (* *p* = 0.0178), miR-29a-5p (*p* = 0.1331), miR-376a-3p (** *p* = 0.0084), miR-26a-5p (*p* = 0.2709), let-7g-5p (*p* = 0.7410), miR-15b-5p (*p* = 0.2186), miR-142-3p (*p* = 0.3050), miR-374a-5p (*p* = 0.1854), let-7b-5p (*p* = 0.2405).

**Figure 2 genes-12-00176-f002:**
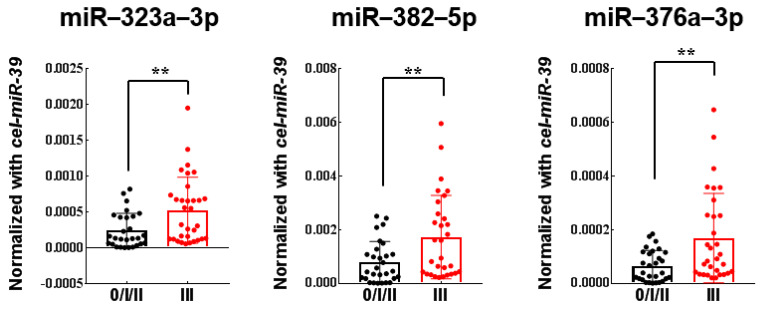
Analysis of 79 plasma samples by qRT-PCR (validation cohort, stage 0/I/IIA vs. IIIB/IIIC). Higher expression of three miRNAs (miR-323a-3p, miR-382-5p, miR-376a-3p) was significantly associated with lymph node metastasis. miR-323a-3p (** *p* = 0.0049), miR-382-5p (** *p* = 0.0053), miR-376a-3p (** *p* = 0.0024).

**Figure 3 genes-12-00176-f003:**
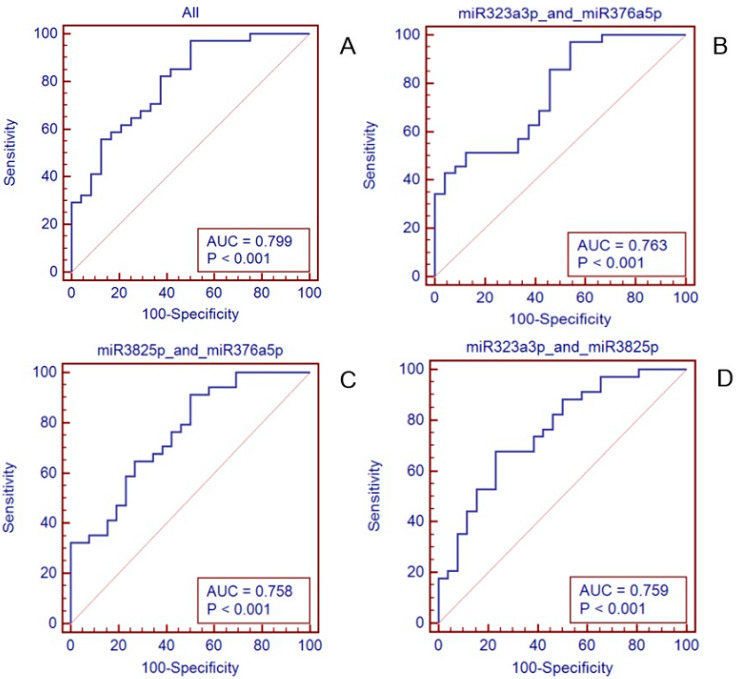
Predictive accuracy of the 3 miRNAs & 2 miRNAs to distinguish patients with LN metastasis from those without LN metastasis. (**A**) 3-microRNA signature (miR-323a-3p, miR-382-5p, miR-376a-3p). (**B**) 2-microRNA signature (miR-323a-3p, miR-376a-3p). (**C**) 2-microRNA signature (miR-323a-3p, miR-382-5p). (**D**) 2-microRNA signature (miR-382-5p, miR-376a-3p).

**Figure 4 genes-12-00176-f004:**
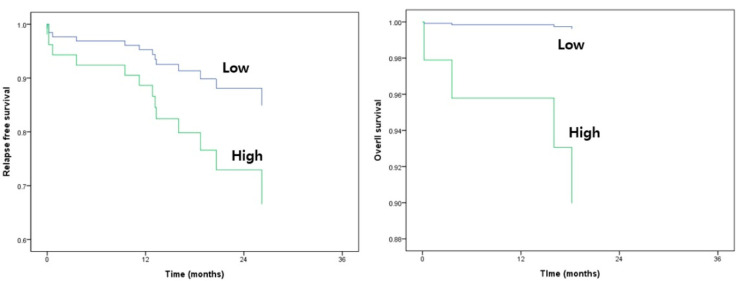
Survival curves for relapse-free survival (RFS) and overall survival (OS) according to three miRNA expression.

**Table 1 genes-12-00176-t001:** Patient characteristics.

Factors	*n* = 79
Age	63
Median, range	(37–86)
Age	
<65	45 (57)
≥65	34 (43)
Gender	
Male	47 (59.5)
Female	32 (40.5)
Tumor location	
Colon cancer	76 (96.2)
Rectal cancer	3 (3.8)
Pathologic T stage	
Tis	1 (1.3)
T1	6 (7.6)
T2	3 (3.8)
T3	53 (67.1)
T4	16 (20.3)
Pathologic N stage	
N0	38 (48.1)
N1	20 (25.3)
N2	21 (26.6)
Pathologic Stage	
I	9 (4.5)
II	28 (35.4)
III	39 (49.4)
IV	3 (3.8)
Differentiation	
well	3 (3.8)
moderate	53 (67.1)
poor	18 (22.8)
others	3 (3.8)
KRAS	
wild	42 (53.2)
mutant	30 (38)
MSI	
low	69 (87.3)
high	6 (7.6)
Histology	
adenocarcinoma	78 (98.7)
others	1 (1.3)
Lymphovascular invasion	
present	43 (54.4)
absent	35 (44.3)
Venous invasion	
present	43 (54.4)
absent	32 (40.5)
Perineural invasion	
present	54 (68.4)
absent	24 (30.4)
lymphocyte response	
present	57 (72.2)
absent	18 (22.8)
Recurrence	16 (20.3)
Death	4 (5.1)

**Table 2 genes-12-00176-t002:** Fold changes and associated *p*-value for 10 miRNAs in the discovery cohort.

Probe Name	Fold Change	Pstage IIIC	Pstage 0 + 1 + IIA	*p*-Value
hsa-miR-323a-3p	2.46	50.15	20.42	0.00616289
hsa-miR-382-5p	1.74	50.05	28.74	0.03742823
hsa-miR-29a-3p	1.73	63.28	36.54	0.01971120
hsa-miR-376a-3p	1.68	78.24	46.48	0.04374451
hsa-let-7b-5p	−1.37	215.72	294.92	0.03424014
hsa-miR-374a-5p	−1.71	25.16	42.91	0.04004743
hsa-miR-142-3p	−1.91	52.04	99.52	0.01377911
hsa-miR-15b-5p	−2.00	52.78	105.47	0.04035384
hsa-let-7g-5p	−2.08	41.97	87.28	0.02688495
hsa-miR-26a-5p	−2.73	17.93	49.02	0.00180513

**Table 3 genes-12-00176-t003:** Univariate and multivariate analysis of overall survival.

	Relapse-Free Survival			Overall Survival		
Variable	Univariate Analysis	Multivariate Analysis		Univariate Analysis	Multivariate Analysis	
	*p*-Value	HR (95% CI)	*p*-Value	*p*-Value	HR (95% CI)	*p*-Value
Sex (M)	0.786		0.777	0.777		
Age	0.744		0.346	0.346		
Pstage (III and IV)	0.038 *	10.859 (0.963–122.475)	0.317	0.317		
histologic type (2)	0.733		0.862	0.862		
histologic grade (2)	0.714		0.976	0.976		
histologic grade (3)	0.749		0.975	0.975		
histologic grade (4)	0.984		1.000	1.000		
LVI (1)	0.075		0.392	0.392		
Venous invasion (1)	0.034 *	2.013 (0.507–7.985)	0.443	0.443		
PNI (1)	0.090			0.428		
miR323a3p×1000 (>0.1326 §)	0.258	0.745 (0.142–3.906)	0.728	0.405	487,477.61 (0.000–∞)	0.974
miR3825p×1000 (>2.1516 §)	0.132	9.399 (0.499–177.143)	0.135	0.961	5.016 (0.517–48.628)	0.164
miR376a5p×1000 (>0.1454 §)	0.251	0.231 (0.010–5.319)	0.360	0.471	0.00 (0.000–2.31 × 10^302^)	0.969

*: Statistically significant with *p* < 0.05. §: Cut off value was obtained by ROC analysis.

## Data Availability

The data presented in this study are available on request from the corresponding author. The data are not publicly available due to privacy.

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
