# Peer review of "Predictive Value of Circulating miRNAs in Lymph Node Metastasis for Colon Cancer"

_genes, 2021, doi:10.3390/genes12020176_

Round 1
Reviewer 1 Report
The manuscript examines circulating miRNA that are disregulated in metastatic colon cancer to identify candidate biomarkers that could predict LN metastasis. Overall, the manuscript is well written and clear.
A few notes:
- The manuscript could benefit from examining the known and/or predicted mRNA targets for miRs-323a-3p, 382-5p and 376a-3p and their involvement in colon cancer and metastasis.
- Typo line 71: exit
- Table one should specify that it is outlining the validation cohort
Reviewer 2 Report
The topic of this manuscript is interesting, but I suggest authors to revise it accuratly.
here, some my comments:
- How did authors calculate relative expression using 2-DDct, without having normal group?
- Authors say to find 10 significantly dis-regulated miRNAs (4 up-regulated and 6 down-regulated miRNAs) in the discovery cohort, as shown in Table 1; but then in Figure 1 the 6 down-regulated miRNAs and the up-regulated miR-29a are indicated as not significantly differentially expressed between the LN-negative and the LN-positive groups in the same cohort. How is it possible?
- Line 150: should be Figure 1 and not Figure 2.
- Line 165; should be described the caption of figure 1 and not table 4. Here again, appears the discrepancy above observed (see point 2)
- Before to show ROC curve analysis of 3 or two miRNAs, authors should show ROC curve analysis for each of 3 miRNA identified as differentially expressed in the validation cohort.
Reviewer 3 Report
Compliments to the authors for an interesting study. In general, it was well conducted, the results support the aim of the study, materials and methods have been described properly. However, there is some room for improvement: 1. l.51 - I suggest a change for "lymph node metastases" instead of "presence of metastatic tumors in lymph nodes" 2. l.61 - hardly readable - explain the reference to TNM? 3. l. 63 - 25% of LN (-) patients develop recurrence - yes, mainly due to poor quality of surgery (insufficient LN harvest) 4. please expand the limitations section 5. what was the median lymph node harvest in the resected specimens? What was the R0 resection rate? Please provide more strictly surgical data 6. How many patients received adjuvant treatment? Is this of any relevance to survival?
